# Contemporaneous symptom networks for multidimensional symptom experience in lung cancer survivors of immunotherapy: A network analysis

Hang Gao[1]☯, Xiaoxue Wen[2]☯, Xiaoli Sun[1], Yuwei Liu[3], Jianwen Hou[4], Danwen Zheng[1], Hejia Chen[1], Lingfang Ma[1], Yufang Zhou[1], Xinyan Yu[5]*

1 Phase I Clinical Ward, Zhejiang Cancer Hospital, Hangzhou, Zhejiang Province, China, 2 Department of Colorectal Medicine, Zhejiang Cancer Hospital, Hangzhou, Zhejiang Province, China, 3 Department of Thoracic Medicine, Zhejiang Cancer Hospital, Hangzhou, Zhejiang Province, China, 4 Head and Neck Surgery, Zhejiang Cancer Hospital, Hangzhou, Zhejiang Province, China, 5 Postgraduate training base Alliance of Wenzhou Medical University (Zhejiang Cancer Hospital), Hangzhou, Zhejiang, China

☯ These authors contributed equally to this work.
* yuxinyansafe@163.com

## Abstract

### Background

Immunotherapy dramatically increases patient survival and radically changes the way lung cancer is treated. Through intricate network analysis of disease-related symptoms, symptom networks are able to statistically analyze and depict the links between different symptoms. Through symptom network analysis, this study seeks to pinpoint the primary symptoms that immunotherapy-treated lung cancer survivors encounter. Furthermore, by leveraging the synergistic relationships between symptoms, it aims to investigate the target actions for precise interventions, offering important insights for the creation of a successful symptom management program.

### Methods

We assessed the symptoms of 249 lung cancer patients undergoing immunotherapy at Zhejiang Cancer Hospital between February and October of 2024. The evaluation was conducted using the Lung Cancer-Specific Module and the Anderson Symptom Assessment Scale (Chinese version). Following the use of exploratory factor analysis to discover symptom clusters, we calculated the centrality indices and created a network structure using the R programming language that displayed the connections between the symptoms. After controlling for factors, we constructed contemporaneous networks that had all 17 symptoms.

**Data availability statement:** All relevant data are within the paper and its Supporting Information files.

**Funding:** This study was supported by a grant from the 2023 Zhejiang Province Medical and Health Science and Technology Program (No. 2023KY568). There was no additional external funding received for this study.

**Competing interests:** The authors have declared that no competing interests exist.

## Results

Four symptom clusters—respiratory, emotional, gastrointestinal, and neuro-perceptual—were shown to be generalized. The three most prevalent symptoms, as determined by nodal intensity, were sadness (rs = 7.43), cough (rs = 6.65) and nausea (rs = 6.73). The most common symptoms at bridge intensity were nausea (rs = 4.69), cough (rs = 4.72), and sadness (rs = 5.69). The network's overall strength and structure did not significantly differ between the male and female groups, or between those who had or did not have a history of smoking.

## Conclusion

This study demonstrates that the symptom burden is significant among survivors of lung cancer immunotherapy, with sadness, cough, and nausea playing crucial roles in the multidimensional symptom network. Interventions focused on addressing sadness can effectively reduce the severity of the entire symptom network, while early intervention for cough and nausea can alleviate the symptom management burden for patients. Additionally, identifying more predictable symptoms can aid in selecting appropriate targets for symptom management. Healthcare professionals can utilize these symptom patterns to deliver evidence-based and precise symptom management for survivors of lung cancer immunotherapy.

## Introduction

Lung cancer is an important public health problem that affects human health worldwide, and there are two main types: non-small-cell lung cancer (NSCLC) and small cell lung cancer (SCLC). Histologically, NSCLC is the most common subtype of lung cancer, accounting for approximately 85% of all lung cancer diagnoses, including adenocarcinoma and squamous cell carcinoma (SCLC) [1]. According to the latest GLOBOCAN Global Tumor Epidemiology Analysis, lung cancer is at the top of the cancer disease spectrum worldwide in terms of incidence and mortality, accounting for 12.4% and 18.7%, respectively [2].In 2022, there will be about 871,000 new cases of lung cancer and 767,000 new deaths in China, and the incidence and mortality rates are both on the rise with age [3]. Lung cancer patients generally suffer from a multidimensional symptom burden, with fatigue (72–89%), dyspnea (58–76%), pain (52–68%), and psychological distress as the main symptoms, significantly affecting treatment adherence and quality of life [4–6]. Longitudinal studies have shown that symptom peaks in postoperative adjuvant therapy patients are synchronized with the chemotherapy cycle, with cough (83% incidence), dry mouth (76%), and sleep disturbances (68%) being most prominent before surgery [7]. During radiation therapy, fatigue, shortness of breath, and coughing up sputum are the main challenges [8]. The severity of symptoms in lung cancer patients is moderated by a variety of factors such as age, gender, and economic income. Studies have shown that the prevalence of digestive and mood symptom clusters was 32% higher in female patients

compared to males, the risk of mood psychological symptoms was increased by 45% in the low-income group, and the VAS scores of fatigue and sleep disturbances in the older patients (>65 years of age) were 1.8 points higher than those in the younger group [9].Despite the efficacy of treatments such as conventional chemotherapy and radiotherapy, the survival rate of NSCLC is extremely low, with a 5-year survival rate of only about 4 percent [10].

In recent years, immune checkpoint inhibitor (ICI) has changed the status quo of lung cancer treatment, compared with the era of chemotherapy, ICI significantly prolongs the survival time of lung cancer patients, and has become the locally advanced and non-small cell lung cancer (non-small cell lung cancer (NSCLC)) patients' Treatment of choice [11,12]. However, despite the considerable clinical efficacy brought by ICIs, clinical observations have shown that a significant proportion of patients receiving immunotherapy develop a variety of toxic side effects, collectively referred to as immune-related adverse events (irAEs) [13]. IrAEs have a very high prevalence and are characterized by a complexity and variety of symptoms, with unpredictable timing and site of manifestation, and have the potential to affect any organ or system [14]. Although most irAEs are usually mild, severe irAEs can compromise treatment outcomes or even lead to patient death [15]. Therefore, it is imperative to elucidate the associated adverse effects that occur in lung cancer immunotherapy survivors and the symptomatic burden endured.

In 2001, Dodd et al. [16] introduced the concept of symptom clusters, positing that these clusters consist of three or more interrelated symptoms that occur simultaneously. They argued that each symptom within a cluster not only coexists with but also influences the others, forming a stable group of related symptoms that may share similar or distinct etiological mechanisms. Subsequently. In 2010, Molassiotis et al. [17] refined this definition, characterizing a symptom cluster as a collection of two or more clinically relevant symptoms that are interconnected at a specific point in time, while still retaining their unique characteristics. Despite these contributions, a universally accepted definition of symptom clusters remains elusive. It has been proposed that targeted interventions for symptom clusters could enhance the efficacy of symptom management, optimize the utilization of healthcare resources, and address the comprehensive care needs of patients [18]. Prior research has indicated a stage-specific agreement for the investigation of the dynamic evolutionary pattern of symptom clusters associated with lung cancer treatment and their clinical management approaches. The pain-fatigue-sleep disorders-distress tetralogy (68–82%) was the most prevalent symptom cluster in the perioperative stage, according to treatment stage-specific analysis, and its pathophysiological basis might be linked to a systemic inflammatory response brought on by surgical trauma [19,20];The dyspnea-fatigue-anxiety triangle predominates during radiotherapy, and V/Q dysregulation due to radiological lung injury forms a bidirectional modulation with vagal-mediated anxiety [21];The chemotherapy phase differentiated a three-dimensional symptom cluster of lung cancer-specific (hemoptysis/chest pain), psychological (depression/cognitive deficits), and nutritional (cachexia/abnormal taste) symptoms, in which platinum-based chemotherapeutic regimens induced activation of TRPV1 channels leading to the neuropathic pain characteristic of the symptom cluster [22];During the synchronized treatment period, a cluster of radiation pneumonitis symptoms coexist with a pain-pharyngeal pain dichotomy, which is caused by cytokine storm and disruption of the mucosal barrier [23]. Longitudinal investigations have also demonstrated the dynamic evolution of symptom clusters in individuals with lung cancer, with significant remission occurring at 6 weeks and peak symptom severity occurring 3 weeks following radiation therapy [24]. A prospective cohort study by the Gift team (N = 358) confirmed that multidimensional symptom clustering resulted in a 42% reduction in severity scores at 6 months of follow-up, suggesting that the acute phase (8−12 weeks after treatment initiation) is the optimal window for intervention [25]. However, existing studies have dual limitations: Traditional symptom clustering analysis overly relies on statistical co-occurrence and ignores non-linear interactions between symptoms compared to isolated symptoms; the unique symptom network triggered by immune checkpoint inhibitor therapy has not been systematically resolved, and its characteristics may show phase-variable patterns depending on the degree of PD-1/CTLA-4 inhibition. Establishing a new paradigm for symptom cluster research based on network pharmacology and dynamic system modeling will be a key path to breaking through the current management bottleneck.

In recent years, there has been an increasing focus on symptomatic network analytics that utilize machine learning techniques, representing a novel data-driven visual analytics methodology that operates independently of preconceived assumptions. Originating from social network analysis, this approach is adept at estimating intricate relational patterns and elucidating the fundamental characteristics of a network through visualization [26]. Network analysis offers a distinctive perspective that emphasizes the individuality of each symptom alongside their inherent interrelations, proving effective in discerning interactions both among symptoms and between symptom clusters within a network. These interactions can be conceptualized as a network, wherein symptoms are represented as nodes and the associations between them are depicted as edges [27]. The connectivity among various nodes illustrates the complex relational dynamics, while the strength of these connections indicates the degree of proximity in their relationships. Furthermore, metrics such as strength, closeness, and betweenness are employed to assess the centrality of a node; a higher centrality signifies greater importance within the network [28]. Certain symptoms may assume a pivotal role in the network's existence and evolution, referred to as high centrality nodes or "core symptoms," which denote the influence of a symptom on other symptoms and its impact on the overall activation of the network [29]. Network analysis, as an important methodological tool for complex systems research, has been extended from the traditional field of psychopathology [30] to the study of chronic disease symptomatology in recent years, and has shown unique parsing value especially in the fields of HIV/AIDS [31] and malignant tumors [32] domains showing unique parsing value. The method can systematically reveal the dynamic interplay mechanism of multidimensional symptom clusters and their clinical evolution by constructing symptom interaction networks [33].Currently, there are few studies on network analysis in the field of lung cancer, and the existing research mainly focuses on the symptoms of patients during the perioperative period and chemotherapy. Luo et al. [34] investigated 1,255 lung cancer patients undergoing chemotherapy and identified four symptom clusters: fatigue, gastrointestinal, psychoneurological, and respiratory. Their central symptoms were fatigue, vomiting, distress, and hemoptysis, respectively, with bridge symptoms identified as pain, vomiting, dry mouth, and shortness of breath. Wang et al. [35] conducted a survey using the Chinese version of the M.D. Anderson Symptom Inventory (MDASI) among 301 perioperative lung cancer patients. They found that the most prevalent symptoms during the perioperative period were pain, fatigue, and shortness of breath, with shortness of breath identified as the core symptom. The study recommended that healthcare providers prioritize interventions targeting this core symptom while also focusing on pain and fatigue to enhance the efficiency and precision of symptom management in perioperative lung cancer care. In the future, it will be essential to integrate machine learning with multimodal data (e.g., biomarkers, radiomics) to develop real-time monitoring tools, validate precision management strategies through dynamic network-driven intervention trials, and promote innovation in lung cancer symptom management via interdisciplinary collaboration.

The mechanism of heterogeneity of immune checkpoint inhibitor (ICI) treatment response in lung cancer and its difference in clinical manifestations is an important direction of current medical research. Among them, smoking status and gender differences, as key biological variables, modulate the interaction pattern between the immune microenvironment and the clinical symptom network through unique molecular mechanisms. From a pathophysiological perspective, smoking-induced bi-directional immunomodulation is of particular interest: tobacco combustion products (e.g., benzo(a) pyrene), through activation of the aromatic hydrocarbon receptor (AhR) signaling pathway, can promote the up-regulation of programmed death ligand 1 (PD-L1) expression and the recruitment of myeloid-derived suppressor cells (MDSCs) to form an immunosuppressive micro environment [36]. This paradoxical effect may lead to a unique symptom interaction characterized by smoking patients – both a treatment response advantage due to elevated TMB and a synergistic worsening of respiratory-consumptive symptom clusters (coughing, shortness of breath, weight loss) exacerbated by chronic inflammation and comorbidities (e.g., COPD). In the gender difference dimension, the sex hormone-immunity axis regulatory mechanisms showed significant biological specificity. Estrogens enhance the antitumor activity of CD8 + T cells by modulating their mitochondrial metabolic reprogramming [37], whereas androgens reduce tissue repair capacity by inhibiting M2 polarization of tumor-associated macrophages (TAMs). This immunomodulatory difference may directly contribute to the

gender dimorphism in symptom characterization. patients are more likely to present with fatigue-mood symptom clusters (e.g., fatigue, sadness, sleep disturbances), whereas somatic symptoms dominate men. A recent meta-analysis further confirmed that women exhibit superior pathologic remission rates in ICI monotherapy or combination therapy(major pathological responses(mPR): OR=1.82, 95%CI 1.13-2.93, p=0.01; pathological complete responses(pCR): OR=1.65, 95%CI 0.97-2.75, p=0.08) [37].These biological differences ultimately map to the topology of the clinical symptom network: smokers exhibit strong respiratory-metabolic symptom associations due to airway inflammation and a predisposition to systemic malignancy, while women have closed-loop feedback for emotional-somatic symptoms due to neuroendocrine-immune interactions, and this heterogeneity of the symptom network requires that healthcare providers implement precise symptom management strategies.

## Methods

### Study design and participants

This cross-sectional study was carried out at Zhejiang Cancer Hospital between February and October of 2024. The start of the recruitment period for this study is on 8/2/2024, and the end of the recruitment period for this study is on 31/10/2024. Participants were deemed eligible if they satisfied the following criteria: (1) a pathological diagnosis of lung cancer, with or without solitary metastases, (2) an age of 18 years or older, (3) a history of receiving immunotherapy for a duration exceeding one month, and (4) voluntary consent to participate in the survey. Exclusion criteria encompassed individuals diagnosed with other malignancies, as well as those demonstrating impaired communication abilities, cognitive deficits, or mental health disorders.

The sample size for this study was calculated using the cross-sectional sample size formula $n=[Z\alpha/2(1-P)P)]/\delta2$, where α was set at 0.05 and the permissible error (δ) was also established at 0.05. A pre-test indicated a symptom incidence rate of P=0.76. Taking into account a projected 20% attrition rate, a minimum of 172 cases was necessary for inclusion in the sample; ultimately, a total of 249 cases were incorporated into the study. All included participants gave informed consent to voluntarily participate in the study and signed a paper version of the informed consent form. Data collection was conducted through face-to-face interviews by two researchers, and all questionnaires underwent independent quality checks. This research adhered to the principles outlined in the Declaration of Helsinki and received approval from the Medical Ethics Committee of Zhejiang Provincial Cancer Hospital, under ethical approval number IRB-2024–149.

### Tools

**General information questionnaire.** The research team developed a comprehensive questionnaire aimed at gathering sociodemographic and clinical data. This instrument included inquiries regarding gender, age, body mass index, educational attainment, tumor staging, pathological classification, smoking history, monthly family income, and place of residence.

**The Anderson Symptom Assessment Scale (Chinese version).** The Anderson Symptom Assessment Scale (Chinese version) [38] was employed to evaluate the prevalence and severity of symptoms experienced by lung cancer survivors who had undergone immunotherapy within the preceding 24 hours. This scale comprises two subscales: Part 1, the symptom subscale, includes a total of 19 symptoms, with the initial 13 items representing the core components of the scale, while the subsequent 6 items are tailored specifically for Chinese patients with localized, post-treatment lung cancer. The scoring system ranges from 0 to 10, where a score of 0 indicates the absence of symptoms and a score of 10 signifies the most severe manifestation of symptoms; thus, higher scores correlate with greater symptom intensity and severity. Part 2 of the scale consists of 6 items that primarily assess the degree to which these symptoms disrupt daily activities. The reliability of the revised lung cancer module is indicated by a Cronbach's alpha coefficient of 0.773, while the MDASI demonstrates a coefficient of 0.914. The combined total for both scales yields a Cronbach's alpha of 0.922 [38]. In this study, the Cronbach's alpha coefficient for the scale was determined to be 0.883.

**Timing of symptom assessment.** This study used a cross-sectional design, with data collected within $30 \pm 7$ days after patients completed immune checkpoint inhibitor (ICI) therapy. This time window was chosen for two reasons: first, the timing of immune-related adverse effects is unpredictable (median ranging from 1 to 6 months) [39], and irAEs may continue to develop beyond the end of treatment; and second, this period captures the peak symptom load at the beginning of the end of treatment [40].

## Statistical analyses

Statistical analyses were conducted utilizing SPSS version 24.0 and R version 4.3.0. In the initial phase of descriptive analysis, continuous variables were characterized by their mean and standard deviation, while categorical variables were represented by the frequency of cases and percentage distribution. To identify and control for potential confounding variables that may influence symptoms within the network, multiple linear regression analysis was employed. Patient symptoms were grouped through principal component analysis, and exploratory factors were derived from the common factors using maximum variance rotation. To ensure the accuracy and stability of the symptom network constructed, symptom clusters were extracted based on the following criteria: a symptom prevalence of at least 20%, a factor loading greater than 0.5, and the presence of more than one factor.

A network analysis of the symptoms included in the study was conducted utilizing the R software version 4.3.1. The qgraph package was employed, leveraging the EBICglasso function alongside Spearman correlation analysis to construct the symptom network graph. The graph module and Spearman correlation analysis facilitated the examination of inter-symptomatic relationships, while the Fruchterman-Reingold force-directed layout was utilized to position the most strongly correlated nodes centrally within the graph. The centrality index, specifically the strength measure, was employed to quantify the total number of direct connections between individual symptoms and other symptoms; a higher value on this index indicates a greater potential for a symptom to influence others. The width of the edges in the graph represents the strength of the correlations, while the color coding indicates the direction of the correlations, with green denoting positive correlations and red indicating negative correlations. Closeness centrality reflects the inverse of the distance from a symptom to other symptoms; a higher value suggests that the symptom is more proximate to others and, thus, more likely to occupy a central position within the network. Betweenness centrality measures the number of shortest paths that pass through a symptom; a higher value indicates a greater likelihood of the symptom serving as a bridging element within the network. Symptoms exhibiting the highest betweenness coefficients are classified as core symptoms. The strength centrality measure is regarded as more reliable than other centrality indicators, and in this study, the intensity centrality measure was selected as the primary indicator for identifying the most relevant symptoms [41]. The MGM package is employed to ascertain the predictability of individual nodes, which facilitates the assessment of the predictive capacity of all other nodes within the network concerning a specified node [42]. The predictability of a node functions as a criterion for assessing the ability of other nodes within the network to anticipate the behavior of a particular node. A symptom that demonstrates a high predictability value indicates that it can be efficiently regulated by its neighboring nodes. To evaluate the stability of the established network, the botnet package is utilized, and the correlation stability coefficient is calculated, with a minimum acceptable threshold set at 0.25 and an ideal target exceeding 0.5 [43].

In the sensitivity analysis, we used a multilevel confounder control strategy: firstly, the covariates with significance level of $P < 0.001$ in the multiple regression analysis were included in the network model as confounders; then we conducted Network Comparison Test (NCT) based on the Network Comparison Test package in R (version 2.2.1) to assess the heterogeneity of the network topology among different subgroups through 10,000 permutation test. A Network Comparison Test (NCT) was conducted based on the Network Comparison Test package in R (version 2.2.1) to assess the heterogeneity of the network topology among different subgroups through the 10,000 permutation test. The NCT analytical framework consists of an invariance test with three dimensions: 1) Network Structure Invariance: It is assumed that the overall topology of the network (i.e., the connection pattern between nodes) is not significant among different

subgroups and that the overall topology of the network is not significant. Network Structure Invariance: It is assumed that the overall topology of the network (i.e., the connection pattern between nodes) is not significantly different between different subgroups, reflecting the group universality of symptom interaction patterns; 2) Global Strength Invariance: It is assumed that the overall connection strength (i.e., the sum of the absolute values of the weights of all edges) of the network is not significantly different between different subgroups, reflecting the group consistency in the overall loading level of symptom clusters; 3)Edge Strength Invariance. Edge Strength Invariance): It is assumed that the connection strength between specific symptom pairs does not differ significantly across subgroups and is the key symptom interaction pathway for identifying group specificity. The above methodological framework follows the best practice guidelines proposed by van Borkulo et al.(2022) [44], and the statistical robustness of the results is ensured by two-tailed tests and false discovery rate correction. (Symptom Network Intergroup Comparison Codes see Supplementary Document 1: R-language code.)

The value of network analysis as an important methodological tool for complex systems research is not only limited to describing single-dimensional features such as the incidence or severity of symptoms, but also in revealing the dynamic interaction mechanisms among symptoms through topological indicators (e.g., centrality, intensity, tightness, and mediativity) [45]. This advantage is particularly prominent in the study of immunotherapy-related symptoms: as immune-related adverse events (irAEs) often present a subclinical persistence, the traditional dichotomous variable (presence/absence) loses the information about the gradient of symptom intensity, whereas continuous severity scores capture the dynamic evolution of the symptom load with greater precision. Moreover, from the methodological point of view: 1) the correlation between severity score and patients' quality of life was significantly higher than that of the incidence index, suggesting that it has more prognostic predictive value [46]; 2) the biased correlation network, also known as Gaussian graph model, constructed by network analysis requires the input of continuous variables, while the incidence data need to use Logistic network model [28]; And the severity gradient can reflect both the intensity dimension and the time dimension (e.g., fluctuation frequency, duration) of the symptoms, providing a richer basis for decision-making for dynamic intervention. Therefore, the symptom network constructed in this study reflects the covariate pattern of symptom severity rather than the epidemiological distribution characteristics of symptoms. This network analysis framework based on symptom severity can more accurately identify core driving and bridging symptoms and provide a theoretical basis for developing precise symptom management strategies.

## Results

### Characteristics of participants

A total of 286 eligible participants were recruited through a convenience sampling method. However, 37 patients were excluded from the study due to incomplete data, resulting in a final analysis comprising 249 patients, which yielded a valid recall rate of 87.06%. As presented in Table 1, the majority of participants were male and over the age of 60. Notably, 53.8% of the participants had a history of smoking, with an equal distribution of 53.8% residing in urban areas and 46.2% in rural areas. Furthermore, 63.9% of the participants had attained an education level below elementary school. In terms of tumor staging, there was 1 case classified as Stage I, 10 cases as Stage II, 58 cases as Stage III, and 180 cases as Stage IV. Regarding the pathological types of tumors, there were 56 cases of small cell carcinoma, 85 cases of squamous carcinoma, 101 cases of adenocarcinoma, and 7 cases of large cell carcinoma. The body mass index (BMI) distribution indicated that 62 participants were classified as emaciated, 145 as normal weight, 25 as overweight, and 17 as obese. Additionally, 32.5% of the families reported a per capita monthly income within the middle to high-income brackets in China.

Multiple linear regression analyses were conducted with total symptom severity as the dependent variable and participant characteristics as the independent variables. The findings indicated that being female (β = −0.053, P < 0.001), having no history of smoking (β = −0.388, P < 0.001), and possessing a high school education or higher (β = −0.157,

**Table 1. Characteristics of the participants and linear regression analysis of overall symptom severity (n = 249).**

| Variables | Mean (SD) or n (%) | β | P |
|---|---|---|---|
| **Age(years)** | | | |
| <60* | 77 (30.9%) | | |
| ≥60 | 172 (69.1%) | −0.053 | 0.313 |
| **Gender** | | | |
| Male* | 187 (75.1%) | | |
| Female | 62 (24.9%) | 0.386 | <0.001 |
| **Body mass index(kg/m2)** | | | |
| waste away* | 62 (24.9%) | | |
| normal | 145 (58.2%) | 0.075 | 0.218 |
| overweight | 25 (10.0%) | 0.065 | 0.286 |
| obese | 17 (6.8%) | 0.041 | 0.461 |
| **Education level** | | | |
| Primary and below* | 159 (63.9%) | | |
| Secondary schools | 50 (20.1%) | 0.056 | 0.313 |
| High school and above | 40 (16.1%) | −0.157 | 0.004 |
| **Smoking history** | | | |
| No* | 134 (53.8%) | | |
| Yes | 115 (46.2%) | −0.532 | <0.001 |
| **Residence** | | | |
| Urban* | 134 (53.8%) | | |
| rural | 115 (46.2%) | −0.035 | 0.504 |
| **Tumor stage** | | | |
| I* | 1 (0.0%) | | |
| II | 10 (0.4%) | −0.004 | 0.983 |
| III | 58 (23.2%) | −0.267 | 0.441 |
| IV | 180 (72.3%) | −0.229 | 0.527 |
| **Pathological type** | | | |
| small cell carcinoma* | 56 (22.5%) | | |
| squamous carcinoma | 85 (34.1%) | 0.117 | 0.084 |
| Adenocarcinoma | 101 (40.6%) | 0.088 | 0.192 |
| large cell carcinoma | 7 (2.8%) | 0.027 | 0.614 |
| **Family monthly income (yuan)** | | | |
| ≤3000* | 38 (15.3%) | | |
| 3001-6000 | 57 (22.9%) | 0.077 | 0.295 |
| 6001-9000 | 73 (29.3%) | 0.077 | 0.316 |
| ≥9001 | 81 (32.5%) | 0.074 | 0.338 |

Note:*is the reference group,$R^2$ = 0.418,adjusted$R^2$ = 0.373,F = 9.181,$p$ < 0.001.

P = 0.004) significantly influenced overall symptom severity. In the subsequent network analysis, these variables were treated as confounding factors to be controlled. Table 1 displays the outcomes of the linear regression analyses alongside the sociodemographic and clinical characteristics of the participants, which include gender, age, body mass index, education level, tumor stage, pathology type, smoking history, monthly household income, and place of residence.

## Prevalence and severity of symptoms

Table 2 presents the prevalence and severity of various symptoms within the full sample. The aggregate score for the 19 assessed symptoms was recorded at 30.90±15.8. Among the symptoms analyzed, the three most frequently reported were cough (81.9%), nausea (80.7%), and distress (71.9%). Conversely, symptoms exhibiting a prevalence of 20% or less included coughing up blood (15.3%) and lethargy (12.4%). Furthermore, the three symptoms with the highest severity scores were fatigue (2.88±1.66), pain (2.34±1.56), and sadness (2.31±1.73).

## Symptom clusters based on symptom scores

The analysis focused on 17 symptoms exhibiting a prevalence greater than 20%, which yielded a Kaiser-Meyer-Olkin (KMO) statistic of 0.863 and a significant result from Bartlett's test of sphericity ($X^2 = 2524.32$, $P < 0.001$), confirming the appropriateness of conducting factor analysis. Four factors with eigenvalues exceeding 1 were identified, accounting for a cumulative variance of 70.86%. Informed by prior research and consultations with members of the Institute team, the identified symptom clusters were designated as follows: respiratory symptom cluster (including cough, phlegm, chest distress, and shortness of breath), gastrointestinal symptom cluster (comprising nausea, vomiting, weight loss, insomnia, and loss of appetite), emotional symptom cluster (encompassing forgetfulness, sadness, restless sleep, and distress), and neuro-perceptual symptom cluster (featuring numbness, pain, xerostomia, and fatigue). As presented in Table 3, the overall score for the symptom clusters was 30.3±16.26, with a mean score of 1.78±1.51; notably, the neuro-perceptual symptom cluster recorded the highest mean score of 2.13±1.46. The findings from the exploratory factor analysis of the symptom clusters are detailed in Table 3.

## Networks of symptoms based on symptom scores

Utilizing the edge-weighted correlation coefficients derived from the 17 symptoms and 3 confounding variables, a partial correlation network model was developed, as illustrated in Fig 1. Panel A of the Figure presents the symptom network

**Table 2. Prevalence and severity of symptoms(n = 249).**

| Variable of symptom | Number of participants | Prevalence (%) | Severity (0–10) (Mean±SD) |
|---|---|---|---|
| pain | 186 | 74.70% | 2.34±1.56 |
| fatigue | 192 | 77.10% | 2.88±1.66 |
| nausea | 201 | 80.70% | 2.00±1.46 |
| restless sleep | 162 | 65.10% | 1.73±1.63 |
| distress | 179 | 71.90% | 1.78±1.45 |
| shortness of breath | 142 | 57.00% | 1.35±1.68 |
| forgetfulness | 186 | 74.70% | 1.64±1.52 |
| loss of appetite | 176 | 70.70% | 1.58±1.41 |
| drowsiness | 31 | 12.40% | 0.34±1.10 |
| xerostomia | 175 | 66.00% | 1.95±1.33 |
| sadness | 203 | 70.20% | 2.31±1.73 |
| vomiting | 102 | 41.00% | 0.89±1.30 |
| numbness | 170 | 68.30% | 1.69±1.29 |
| cough | 204 | 81.90% | 2.27±1.68 |
| phlegm | 166 | 66.70% | 1.84±1.79 |
| cough with blood | 38 | 15.30% | 0.26±0.64 |
| chest distress | 170 | 68.30% | 1.79+±1.78 |
| insomnia | 145 | 58.20% | 1.16±1.25 |
| weight loss | 152 | 61.0% | 1.09±1.18 |

**Table 3. Symptom clusters based on symptom scores (n = 249).**

| Symptom cluster | Cluster composition | Factor loading | | | | Mean±SD |
|---|---|---|---|---|---|---|
| | | Factor 1 | Factor 2 | Factor 3 | Factor 4 | 1.81±1.73 |
| Respiratory symptom clusters | cough | **0.477** | 0.013 | 0.759 | 0.172 | |
| | phlegm | **0.488** | −0.099 | 0.731 | 0.207 | |
| | chest distress | **−0.010** | 0.039 | 0.141 | 0.284 | |
| | shortness of breath | **0.477** | 0.013 | 0.759 | 0.172 | |
| Gastrointestinal symptom clusters | nausea | 0.758 | **0.342** | −0.009 | −0.087 | 1.34±1.32 |
| | vomiting | 0.388 | **0.742** | −0.140 | 0.045 | |
| | weight loss | 0.421 | **0.639** | −0.065 | 0.103 | |
| | insomnia | 0.452 | **0.725** | −0.163 | −0.050 | |
| | loss of appetite | 0.517 | **0.625** | −0.133 | −0.013 | |
| Emotional symptom clusters | forgetfulness | 0.505 | −0.284 | **0.010** | −0.575 | 1.87±1.58 |
| | sadness | 0.870 | −.142 | **0.046** | −0.135 | |
| | restless sleep | 0.649 | −0.216 | **−0.045** | −0.543 | |
| | distress | 0.663 | −0.234 | **−0.095** | −0.455 | |
| Neuro-perceptual symptom clusters | feeling numbness | 0.618 | −0.328 | −0.351 | **0.412** | 2.13±1.46 |
| | feeling pain | 0.657 | −0.363 | −0.301 | **0.397** | |
| | feeling xerostomia | 0.625 | −0.295 | −0.355 | **0.366** | |
| | feeling fatigued | 0.807 | −0.292 | −0.190 | **0.130** | |

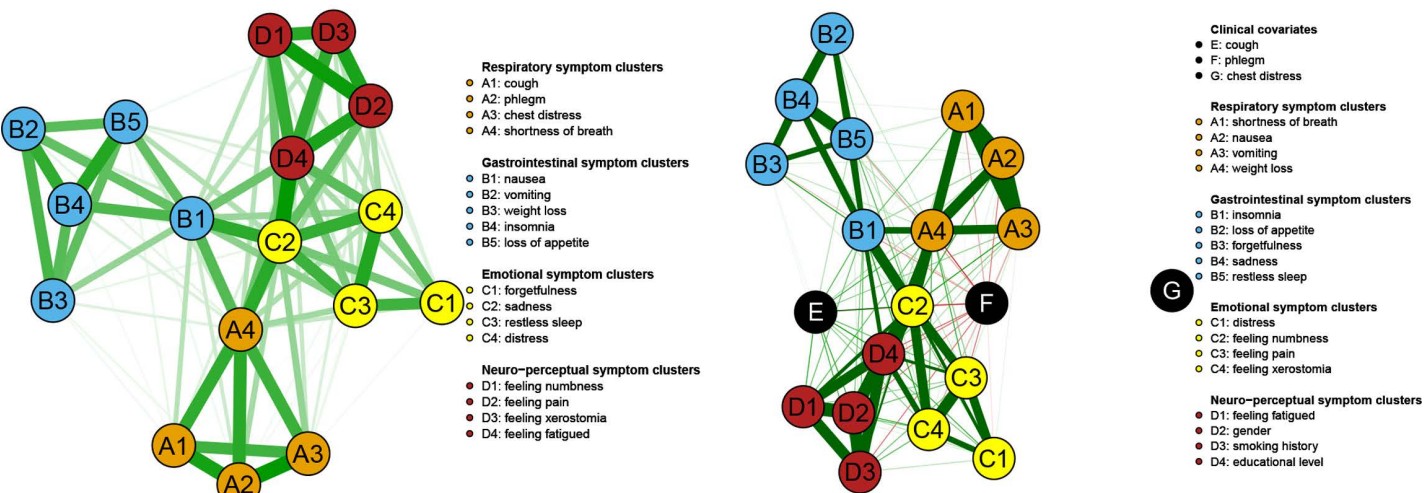

## A.Total without clinical covariates

## B.Total with clinical covariates

Note:The green line represents positive correlation and the line represents

**Fig 1. Symptom networks of the total sample with and without clinical covariates.** A: Total without clinical covariates; B: Total with clinical covariates. (The green line represents positive correlation and the red line represents negative correlation. Respiratory symptom clusters: A1: cough; A2: phlegm; A3: chest distress; A4: shortness of breath; Gastrointestinal symptom clusters: B1: nausea; B2: vomiting; B3: weight loss; B4: insomnia; B5: loss of appetite; Emotional symptom clusters:C1: forgetfulness; C2: sadness; C3: restless sleep; C4: distress; Neuro-perceptual symptom clusters: D1: numbness; D2: pain; D3: xerostomia; D4: fatigue).

devoid of covariates, whereas Panel B depicts the symptom network inclusive of covariates. In this symptom network, the coloration of the edges signifies the type of correlation present among the symptoms; specifically, green edges represent positive correlations, while red edges denote negative correlations.

In the comprehensive network analysis, the three strongest associations identified were between sadness and fatigue (r = 0.76), shortness of breath and chest distress (r = 0.72), and pain and numbness (r = 0.71). When clinical covariates were incorporated into the networks, the strength of each connection diminished; however, the interconnections among symptoms remained largely unchanged. Noteworthy associations were observed between gender and numbness (r = 0.38), gender and fatigue (r = 0.42), and gender and sadness (r = 0.44), as well as between smoking history and cough (r = −0.39) and smoking history and sadness (r = −0.39). Conversely, educational level exhibited only a weak correlation with other symptoms. Detailed information regarding the weights of each connection in the networks, both with and without clinical covariates, can be found in Supplementary Table 1 (Clinical covariate group weights) and Supplementary Table 2 (No clinical covariate group weights). Furthermore, the outcomes of the centrality analyses are illustrated in Fig 2, where Panel A depicts the centrality index of the symptom network absent clinical covariates, and Panel B presents the centrality index of the symptom network inclusive of clinical covariates. In the absence of clinical covariates, the most prominent symptoms based on intensity were sadness (rs = 7.43), nausea (rs = 6.73), and cough (rs = 6.65). In contrast, within the network that included clinical covariates, the leading symptoms were sadness (rs = 8.40), cough (rs = 7.84), and fatigue (rs = 7.52). The strongest correlational associations identified were between shortness of breath and chest distress (r = 0.72), cough and chest distress (r = 0.66), chest distress and phlegm (r = 0.65), cough and sadness (r = 0.64), and insomnia and vomiting (r = 0.63). Importantly, as depicted in Fig 3, which shows bridge strength with and without clinical covariates, sadness emerged as the central bridging element in both networks (rs = 5.69 without clinical covariates vs. rs = 6.66 with clinical covariates).

## A. The centrality index without clinical covariates

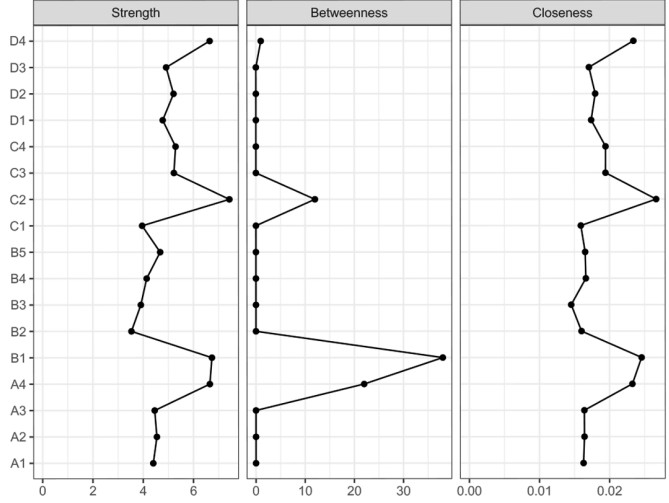

## B. The centrality index with clinical covariates

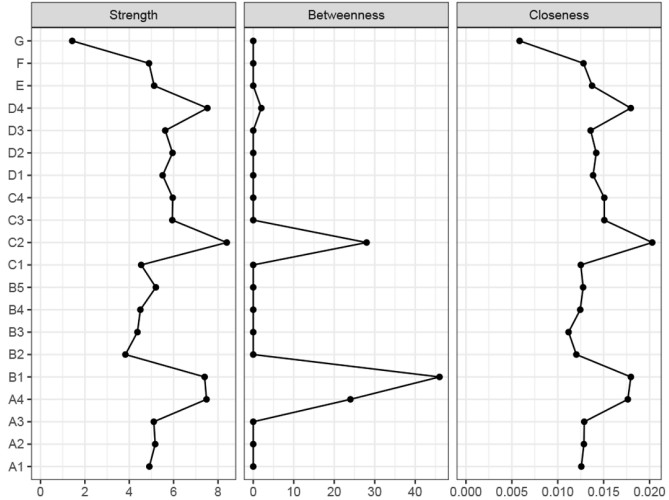

**Fig 2. Centrality index for total sample symptom networks.** A: the centrality index without clinical covariates; B: the centrality index with clinical covariates (Strength centrality: the sum of the weights of all edges of a node; Betweenness centrality: how often a node appears in all the shortest paths in the network; Closeness centrality: the inverse of the average length of the shortest path for a node to reach all other nodes in the network).

**A. The bridge strength without clinical covariates**

**B. The bridge strength with clinical covariates**

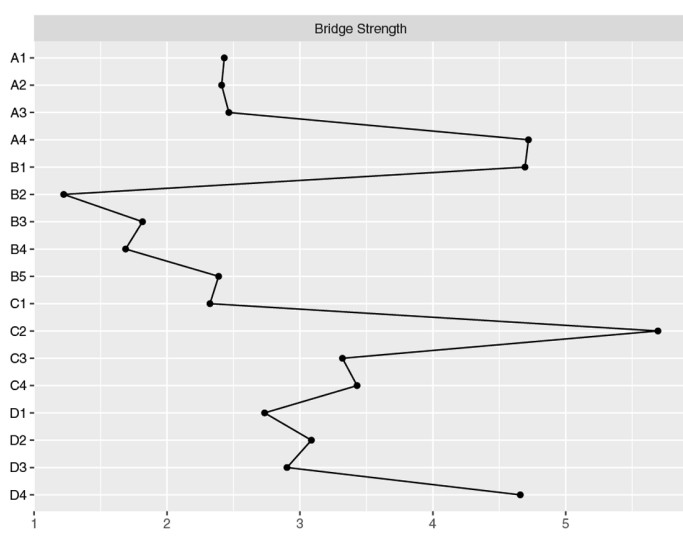
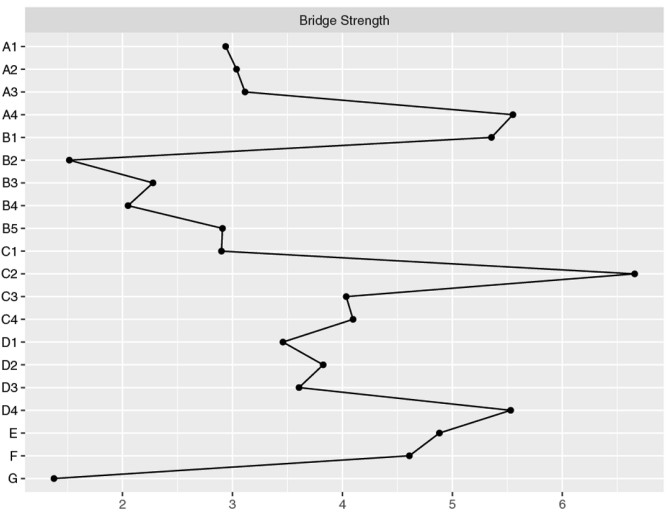

**Fig 3. Bridge Strength for Total Sample Symptom A: The bridge strength without clinical covariates; B: the bridge strength with clinical covariates.** (Bridge Strength: how often a node appears in all the shortest paths in the network).

## Accuracy and stability of the network

The edge-weighted values presented in this sample were found to coincide with the 95% confidence intervals (CIs) of the edge weights derived from the bootstrap method, as illustrated in Fig 4. This overlap suggests that the edge weights possess a satisfactory level of accuracy. Furthermore, the results from the stability test of the symptom network revealed that the correlation stability coefficients for node strength and bridge strength were 0.751 (excluding clinical covariates) and 0.671 (including clinical covariates), respectively. Both coefficients exceed the threshold of 0.5, indicating a robust level of network stability. Additionally, the findings from the centrality difference test, depicted in Fig 5, show that the black squares denote statistically significant differences between the two node strengths along the axes. Notably, the majority of nodes exhibiting higher centrality demonstrate significant differences from those with lower centrality across the two networks, thereby suggesting that the node strengths within the networks are relatively stable.

## Gender-network comparison

The symptom networks for the male and female groups are shown in Fig 6. The NCT results showed no significant difference in the topology of the male and female symptom networks based on the network invariance test (M=0.279, *P*=0.252), as shown in the right panel of Fig 7 and the global connection strengths did not reach significance (S=0.972, *P*=0.101), as shown in the left panel of Fig 7. However, the male group exhibited higher network density (global strength S=7.182 for the male group vs. S=6.210 for the female group). Notably, although there were no overall differences between the symptom networks of the male and female groups, edge-weighting analyses revealed significant differences between the margins of nausea and vomiting (E=0.279, *P*=0.002). Analysis based on the centrality invariance test showed that sadness (male intensity centrality rs=6.790 vs. female intensity centrality rs=5.658) and fatigue (male rs=6.098 vs. female rs=4.960) were core symptoms in both groups.

## Smoking history-network comparison

The network of symptoms with and without a history of smoking is shown in Fig 8. The NCT results showed that based on the network invariance test, there was no significant difference in the topology (M=0.187, *P*=0.793, shown in the

## A.without clinical covariates

## B.with clinical covariates

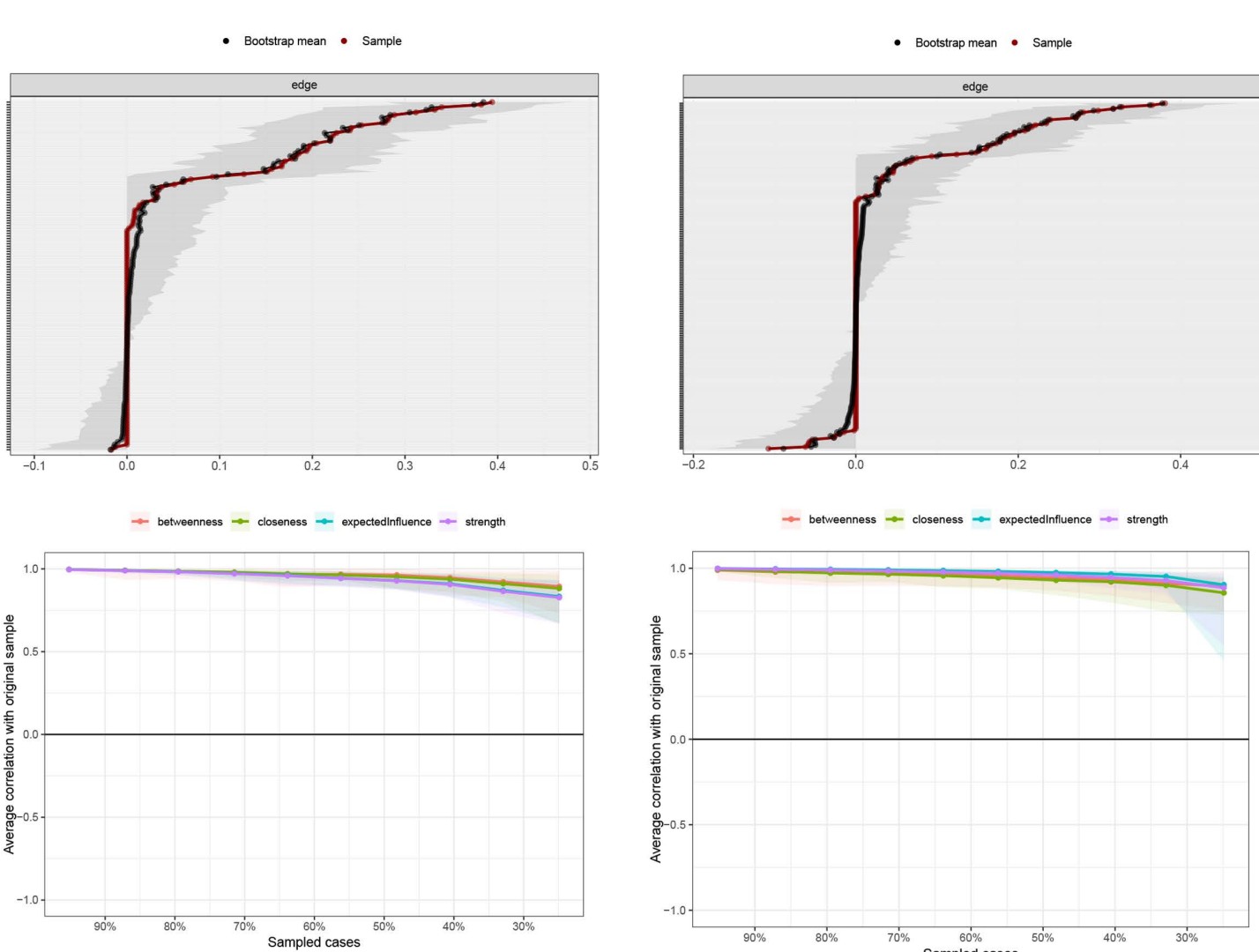

**Fig 4. Accuracy and Stability of Total Sample Symptom A: without clinical covariates; B: with clinical covariates (The top panel is an exactness test for edge weights; the bottom panel is a stable direction test for centrality).**

right panel of Fig 9) and global connection strength (S=0.376, *P*=0.260, shown in the left panel of Fig 9) of symptomatic networks in the smoking group versus the nonsmoking group, but the network was denser in the smoking group (global strength of the smoking group S=7.231 vs. global strength S=6.855) in the non-smoking group. Notably, although there were no overall differences between the symptom networks of the smoking and non-smoking groups, edge weighting analyses revealed a significant difference between shortness of breath vs. weight loss (E=0.187, *P*=0.001), sadness vs. pain (E=0.108, *P*=0.048), vomiting vs. forgetfulness (E=0.091, *P*=0.001), and shortness of breath versus loss of appetite (E=0.035, *P*=0.014) had significant differences in the margins. Analysis based on the centrality test showed that the core symptoms in the smoking group were sadness (intensity centrality rs=6.121), cough (rs=5.787) and nausea (rs=5.768), and in the non-smoking group, the core symptoms were sadness (rs=6.814), vomiting (rs=6.322), and lethargy (rs=6.206).

A.without clinical covariates

B.with clinical covariates

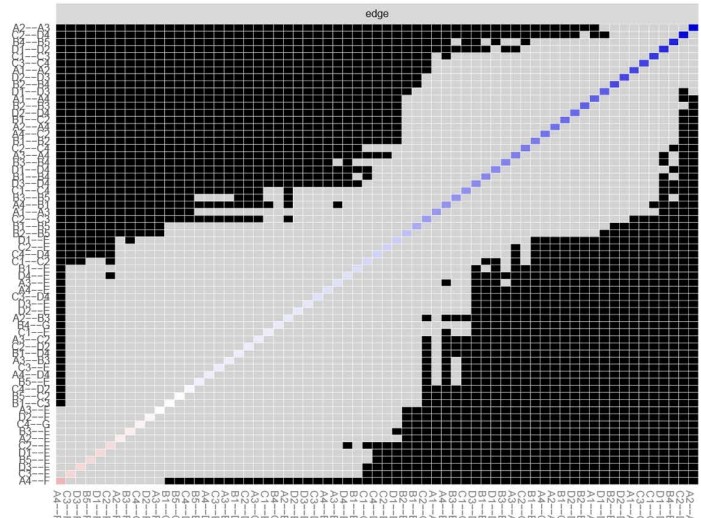

**Fig 5. Bootstrap difference test of edge weight of Total Sample Symptom Networks A: without clinical covariates; B: with clinical covariates.**

A.male(n=187)

B.female(n=62)

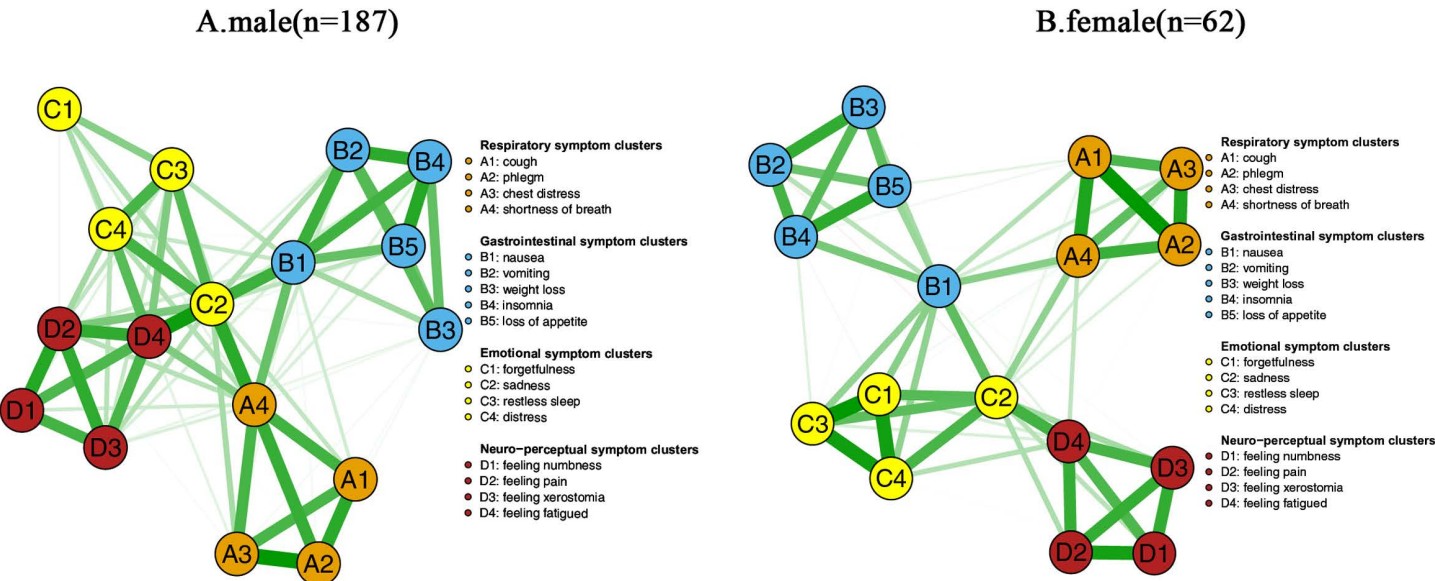

**Fig 6. Gender-specific symptom networks(n = 249) A: male(n = 187); B: female(n = 62) (The green line represents positive correlation and the red line represents negative correlation.** Respiratory symptom clusters: A1: cough; A2: phlegm; A3: chest distress; A4: shortness of breath; Gastrointestinal symptom clusters: B1: nausea; B2: vomiting; B3: weight loss; B4: insomnia; B5: loss of appetite; Emotional symptom clusters:C1: forgetfulness; C2: sadness; C3: restless sleep; C4: distress; Neuro-perceptual symptom clusters: D1: numbness; D2: pain; D3: xerostomia; D4: fatigue).

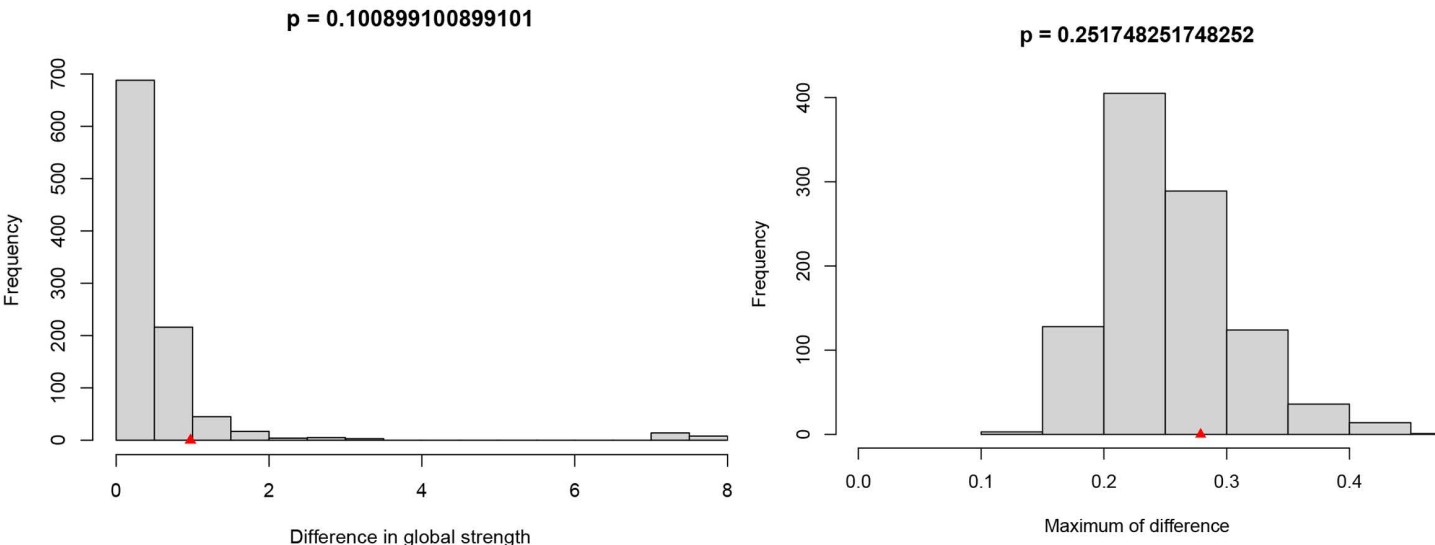

**Fig 7. Comparison of networks properties between females and males** (The left-hand panel shows the results of the global invariance test; The right-hand panel shows the results of the network invariance test).

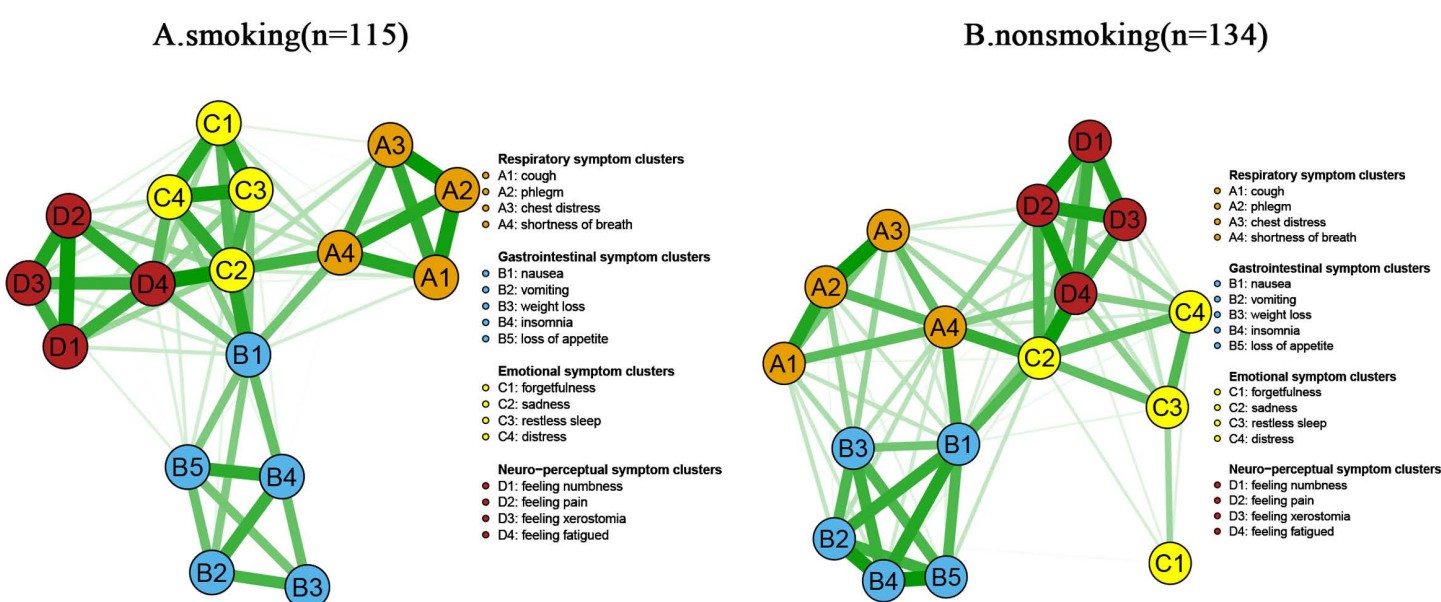

**Fig 8. Smoking-history specific symptom networks(n =249) A: smoking(n=115); B: nonsmoking(n=134)** (The green line represents positive correlation and the red line represents negative correlation. Respiratory symptom clusters: A1: cough; A2: phlegm; A3: chest distress; A4: shortness of breath; Gastrointestinal symptom clusters: B1: nausea; B2: vomiting; B3: weight loss; B4: insomnia; B5: loss of appetite; Emotional symptom clusters:C1: forgetfulness; C2: sadness; C3: restless sleep; C4: distress; Neuro-perceptual symptom clusters: D1: numbness; D2: pain; D3: xerostomia; D4: fatigue).

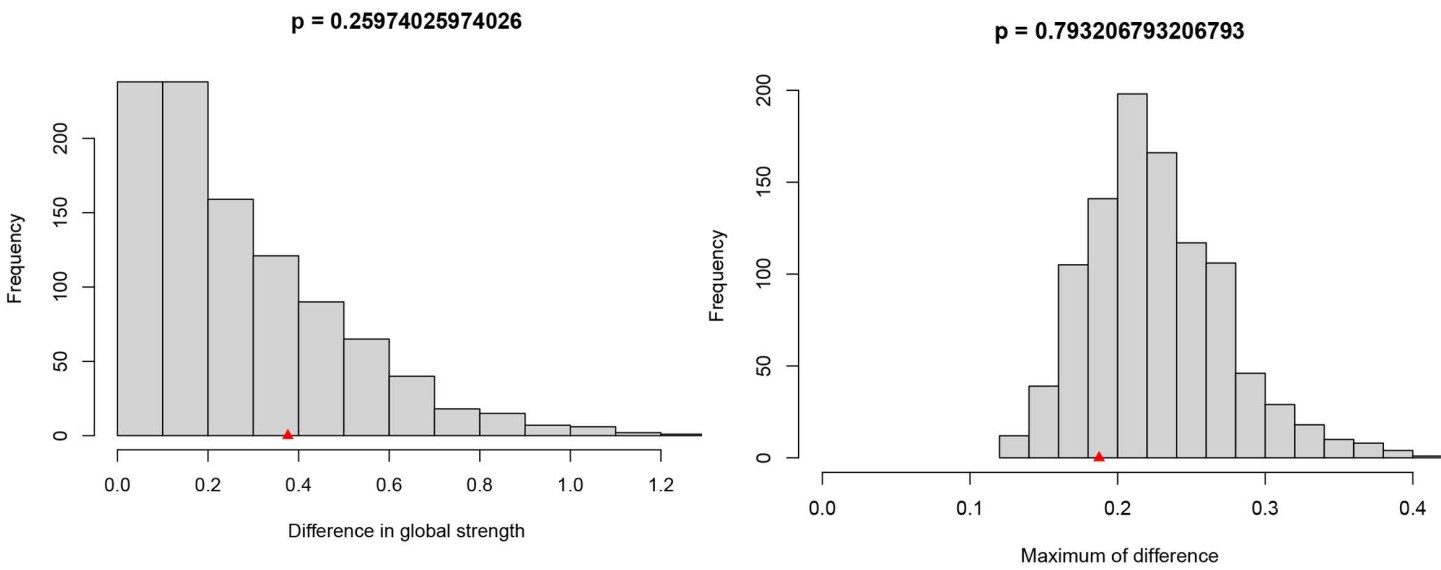

**Fig 9. Comparison of networks properties between with and without smoking-history (The left-hand panel shows the results of the global invariance test; The right-hand panel shows the results of the network invariance test).**

## Discussion

To the best of our knowledge, this study represents the inaugural network analysis focusing on lung cancer survivors undergoing immunotherapy. Our results indicate that sadness emerged as the most central symptom within the centrality index, with fatigue, pain, and sadness identified as the three most prevalent symptoms. Additionally, sadness, cough, and nausea were recognized as bridge symptoms, implying that sadness plays a pivotal role in the onset and exacerbation of other symptoms. Consequently, the formulation of interventions aimed at alleviating symptoms of sadness is essential for mitigating the overall symptom burden experienced by lung cancer survivors receiving immunotherapy.

In our analysis of the symptom network, we delineated four distinct clusters: respiratory symptoms (including cough, phlegm, chest distress, and shortness of breath), gastrointestinal symptoms (comprising nausea, vomiting, weight loss, insomnia, and loss of appetite), emotional symptoms (such as forgetfulness, sadness, restless sleep, and distress), and neuro-perceptual symptoms (including numbness, pain, xerostomia, and fatigue). While immunotherapy represents a novel and promising anti-tumor strategy for lung cancer treatment, there is a paucity of literature addressing the symptom clusters experienced by lung cancer survivors undergoing this form of therapy. Our findings reveal significant differences when compared to the symptom clusters reported in postoperative and post-chemotherapy lung cancer patients.

For instance, Luo et al. [34] identified four distinct symptom clusters among chemotherapy patients diagnosed with lung cancer: fatigue, gastrointestinal, psychoneurological, and respiratory symptoms. They employed the Apriori algorithm alongside the initial onset timing of these symptoms to discern pre-symptoms within each cluster. In a separate study, Li et al. [20] examined 217 postoperative lung cancer patients and delineated seven perioperative symptom clusters: lung cancer-specific, sleep disturbance, nervous system, nutritional, gastrointestinal, psychological, and respiratory symptoms, noting that these clusters evolved. The discrepancies observed between prior studies and the findings of the current investigation may stem from variations in cancer treatment approaches, the criteria for participant inclusion, the tools utilized for measurement, and the analytical methods employed. In our study, we utilized the MD Anderson Symptom Scale Lung Cancer Module to evaluate 19 symptoms. Furthermore, while earlier research has typically posited that symptom clustering arises from a shared underlying factor, network analysis presents a more dynamic perspective, positing that

symptoms cluster due to their interactions with one another [47]. This approach emphasizes the distinctiveness of each symptom and its interrelations, employing visualization techniques to illustrate complex networks that elucidate the connections among symptoms. This methodology not only highlights core symptoms but also aids researchers in identifying the most significant symptoms within the network structure, thereby assisting healthcare providers and researchers in the development of targeted and personalized treatment strategies [27]. Prior research has indicated that fatigue is the predominant symptom experienced by patients with lung cancer, with reported prevalence rates ranging from 70% to 100% [48]. This symptom persists throughout treatment and during subsequent follow-up periods, poses significant challenges in terms of effective management, and has been extensively investigated within the context of cancer survivorship.

Our study showed that fatigue was not a core symptom despite its higher prevalence (77.10%) and severity (2.88±1.66) among lung cancer survivors receiving immunotherapy. In the context of symptom networks, Central symptoms are key pivotal nodes in the symptom network, identified by Strength Centrality metrics. Strength centrality is defined as the sum of the absolute values of the edge weights of a node and its directly connected nodes, and the higher the value of strength centrality, the stronger the direct influence of the symptom in the network, and its fluctuations can be rapidly propagated to the neighboring symptoms through the shortest path [49]. Central symptoms have a dual character: not only are they at the top of both the symptom prevalence and severity distributions, but as high-influence nodes in the network, changes in their state can trigger cascading responses in symptom clusters through directly connected pathways. In this study, sadness (intensity centrality rs=8.40 with clinical covariates vs. intensity centrality rs=7.43 without clinical covariates), as a core node symptom, not only had a high prevalence of 70.2%, but its fluctuation also showed a significant covariation with the neighboring symptoms (e.g., fatigue r=0.76, cough r=0.64, nausea r=0.61, and distress r=0.60). and this correlation was not affected by variables such as gender and smoking history. This finding underscores the importance of sadness as a critical symptom that may exacerbate and trigger other symptoms, aligning with previous research outcomes [50,51]. Sadness is frequently observed in individuals who have survived cancer, and this emotional response can be understood as a sequential psychological process that unfolds during a cancer patient's experience. It manifests across psychological, interpersonal, and socio-cultural dimensions [52], which may hinder the individual's capacity to manage the challenges associated with cancer, including physical symptoms and treatment regimens. The rationale for this analysis may stem from several factors, including the unfavorable prognosis associated with lung cancer patients, as well as the cognitive, emotional, and social responses to the anticipated decline in physical functioning due to the illness. These responses can significantly impact the psychosocial and spiritual aspects of an individual's identity. Additionally, maladaptive coping mechanisms in response to symptomatic distress, the financial burden of expensive treatments, concerns about becoming a burden to family members, reluctance to express grief, diminished family roles, and the fear of abandonment may serve as risk factors and triggers for the onset of depressive symptoms. Despite the prevalence of these emotional challenges, they are frequently inadequately addressed and managed in clinical settings [53]. Sadness, characterized as a predominantly self-referential negative emotion, represents one of the affective manifestations associated with depressive states [54]. Research indicates that the intensity of sadness experienced by patients is influenced by several factors, including age, gender, the specific site of cancer, the treatment environment, and the stage of disease progression. This emotional state is correlated with elevated levels of emotional reactivity, social dysfunction, physical discomfort, and terminal insomnia. Furthermore, persistent sadness and its heightened expression may contribute to the advancement of the disease, hinder effective symptom management, intensify negative emotional states, and potentially elevate the risk of suicidal ideation among patients [54,55]. By focusing on the primary symptom, the intervention can diminish the "targeting" effect of other nodes that were initially associated with it, thereby facilitating the propagation of the intervention to adjacent nodes. This process may ultimately result in the reduction or elimination of additional symptoms. Consequently, early identification and effective management of sadness must be integrated into the long-term care strategies for lung cancer survivors. Addressing sadness as a central intervention can mitigate its spread within the symptom network, thereby lessening the intensity of other symptoms and enhancing the overall efficacy of symptom management. Brief screening instruments, such as

the Preparatory Grief in Advanced Cancer Patients Scale (PGAC) developed by MYSTAKIDOU et al [56] in 2005, can be employed to identify patients experiencing clinically significant cancer-related grief. This scale demonstrates strong internal consistency, evidenced by a Cronbach's alpha of 0.838, with elevated scores reflecting a more severe level of anticipatory grief. Healthcare professionals need to conduct accurate assessments and promptly identify, and intervene in cases of sadness. Additionally, they should foster social support systems (including family, medical, and community support) and provide psychotherapeutic options (such as acceptance and commitment therapy, group therapy, and cognitive restructuring) tailored to the individual patient's preferences and needs. Such measures can facilitate the expression of negative emotions, thereby alleviating sadness and enhancing mental health outcomes [57].

Bridge symptoms are pivotal nodes connecting different symptom clusters or disease subgroups in a network topology, identified by the bridge strength centrality metric. Bridge strength centrality is the sum of the absolute values of the edge weights of a node connecting different modules, with higher values indicating a stronger cross-module propagation of the symptom [58]. High bridge-strength symptoms (e.g., sadness in the present study), although not necessarily the highest in severity, act as cross-module propagation mediators and may trigger cascading deterioration of symptom clusters. In the present study, sadness (bridge strength centrality rs = 6.66 with clinical covariates vs. bridge strength centrality rs = 5.69 without clinical covariates) was the bridge symptom with the highest bridge strength in the symptom network, and the sadness associations encompassed respiratory symptom clusters of coughing, nausea, and fatigue, gastrointestinal symptom clusters and neuroperceptual symptom clusters, with a strong link between them, as illustrated in Fig 1(Symptoms networks of the total sample with and without clinical covariates) of the symptom network. This implies that sadness is not only common to multiple symptom clusters but is also able to connect different symptom clusters. In addition, the symptoms of cough and nausea were also found to have a high bridging effect. Studies [58] have shown that by cutting off bridge symptoms, patients can be stopped from spreading from one disease to another. This is because bridge nodes or edges connect different clusters and have a higher ability to spread information in the network. Therefore, in symptom management for patients at this stage of the disease, sadness can be used as a high-impact intervention target throughout immunotherapy for lung cancer, and symptoms such as cough and nausea can be used as a sub-focus of intervention to develop a precise intervention program to weaken the information dissemination capacity of these bridge symptoms in the network, thus improving the effectiveness of symptom management for patients.

Furthermore, after adjusting for all covariates, we did not find any significant differences between the symptom networks in the various groups; however, network density was significantly higher in the smoking group than in the non-smoking group and significantly higher in the male group than in the female group. No comparable results have been documented in other studies. To distinguish between individuals in various illness courses, network density may be a more sensitive predictor than severity [51]. Long-term HIV infection patients have denser symptomatic networks than recently diagnosed patients, according to previous studies [59]. While there is no discernible difference in the severity of symptoms among HIV-positive people, those with dense networks have worse physical and mental health. According to Zhu et al [51], network density is a better indicator of the long-term outcome of tumor treatment when seen through the lens of symptomatology. To evaluate the trajectory of symptomatic network density in lung cancer immunotherapy survivors and further examine the connection between symptomatic network density and quality of life in these survivors, our research team will keep conducting cohorts and studies with extended follow-up.

## Limitations

There are several restrictions on our investigation. First off, the network only demonstrates partial correlation and does not define causality due to the cross-sectional design and randomized sampling method employed in this study. To better understand the dynamic network of lung cancer immunotherapy survivors and identify which symptoms negatively impact others, the research team will carry out more cohort studies. Second, just a few symptoms were examined in this study using the Anderson Symptom Assessment Scale (Chinese version); Next, research should employ a variety of

measurements to confirm our findings. The results of this study are also not generalizable because it was conducted at a single site; additional research could compare the variations across several centers and geographical areas.

## Conclusion

In this study, a network analysis Gaussian graph model was used to construct the symptom network of lung cancer immunotherapy survivors, and sadness was found to have a dual topological feature: It was both a core node symptom and a key bridge symptom, suggesting that it should be a priority target for multidimensional interventions. Analysis of symptom dimensions revealed that nausea had the highest incidence and that fatigue severity was the most significant. Although subgroup analyses did not reveal differences in network structure by gender and smoking history, males and smokers demonstrated higher network density. This study suggests that multimodal interventions targeting sadness may lead to a reduction in global network strength and simultaneous improvement in respiratory, gastrointestinal, and neuroperceptual symptom clusters. Healthcare professionals should pay attention to the significant impact of negative "sadness" emotions on mental health and physical symptoms in lung cancer immunotherapy survivors. In the future, time series network analysis and Mendelian randomization are needed to dynamically track symptom evolution and validate causal effects, providing evidence-based pathways for precision medicine.

## Supporting information

**Supplementary Table 1. Clinical covariate group weights.**
(XLSX)

**Supplementary Table 2. No clinical covariate group weights.**
(XLSX)

**Supplementary Table 3. Relevant data.**
(XLSX)

**Supplementary Document 1. R-language code(Symptom Network Intergroup Comparison Codes).**
(TXT)

## Acknowledgments

We are truly grateful to all the lung cancer patients who participated in our research. We express our gratitude to every member of the Zhejiang Cancer Hospital personnel who helped with data collection during the study period.

## Author contributions

Data curation: Hang Gao, Jianwen Hou, Danwen Zheng.

Formal analysis: Hang Gao.

Investigation: Hang Gao, Xiaoxue Wen, Xiaoli Sun, Yuwei Liu, Hejia Chen, Lingfang Ma, Yufang Zhou.

Methodology: Hang Gao.

Project administration: Xinyan Yu.

Resources: Yuwei Liu.

Supervision: Xinyan Yu.

Writing – original draft: Hang Gao, Xiaoxue Wen.

Writing – review & editing: Xinyan Yu.

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
