## [Decision Letter · Decision Letter 0]

Dear Dr. Yu,

Thank you for submitting your manuscript to PLOS ONE. After careful consideration, we feel that it has merit but does not fully meet PLOS ONE’s publication criteria as it currently stands. Therefore, we invite you to submit a revised version of the manuscript that addresses the points raised during the review process.

**ACADEMIC EDITOR:**

We look forward to receiving your revised manuscript.

Kind regards,

Jincheng Wang

Academic Editor

PLOS ONE

Journal Requirements:

“his study was supported by a grant from the 2023 Zhejiang Province Medical and Health Science and Technology Program (No. 2023KY568).”

4. In the online submission form you indicate that your data is not available for proprietary reasons and have provided a contact point for accessing this data. Please note that your current contact point is a co-author on this manuscript. According to our Data Policy, the contact point must not be an author on the manuscript and must be an institutional contact, ideally not an individual. Please revise your data statement to a non-author institutional point of contact, such as a data access or ethics committee, and send this to us via return email. Please also include contact information for the third party organization, and please include the full citation of where the data can be found.

“This study was supported by a grant from the 2023 Zhejiang Province Medical and Health Science and Technology Program (No. 2023KY568).”

 “his study was supported by a grant from the 2023 Zhejiang Province Medical and Health Science and Technology Program (No. 2023KY568).”

7. ‘Please include a separate caption for each figure in your manuscript.

Additional Editor Comments (if provided):

This manuscript presents an innovative and clinically relevant analysis of symptom networks among lung cancer patients receiving immunotherapy. The authors have employed sophisticated network analysis methodology to examine symptom clusters and their interactions, with particular attention to identifying central and bridge symptoms. While the research makes a valuable contribution to understanding symptom management in immunotherapy patients, several aspects of the manuscript require revision to enhance its clarity and impact.

The introduction would benefit from a more comprehensive synthesis of existing literature, particularly regarding known symptom patterns in lung cancer and previous applications of network analysis in oncology. The methodology section needs greater precision in describing the timing and nature of symptom assessments, as well as clearer justification for the chosen analytical approaches. Of particular importance is the need to specify whether the network analysis was based on symptom severity or prevalence data, as this distinction has significant implications for interpretation.

The results are generally well-presented, though the subgroup analyses comparing gender and smoking status networks would be strengthened by more detailed statistical reporting. The network visualizations are informative but would benefit from clearer labeling and explanation of the metrics used. The discussion effectively interprets the findings but could more explicitly address the clinical implications of identifying sadness as a central symptom.

The manuscript would benefit from professional English language editing to improve clarity and consistency. Additionally, the authors should consider adding supplementary materials to provide detailed network metrics and methodological details that would enhance reproducibility.

Despite these limitations, this study makes a meaningful contribution to our understanding of symptom interactions in lung cancer immunotherapy patients and provides valuable insights for clinical practice. With appropriate revisions to address these concerns, particularly regarding methodological clarity and statistical reporting, this manuscript would be suitable for publication.

Reviewers' comments:

Reviewer's Responses to Questions

**Comments to the Author**

1. Is the manuscript technically sound, and do the data support the conclusions?

Reviewer #1: Partly

Reviewer #2: Yes

2. Has the statistical analysis been performed appropriately and rigorously?

Reviewer #1: No

Reviewer #2: I Don't Know

3. Have the authors made all data underlying the findings in their manuscript fully available?

Reviewer #1: No

Reviewer #2: Yes

4. Is the manuscript presented in an intelligible fashion and written in standard English?

Reviewer #1: No

Reviewer #2: Yes

Reviewer #1: Very important topic and well written overall. please address the below suggestions

1). Introduction: Add one paragraph regarding what are known in symptoms side effects in lung cancer, what are existing literature in sympotm cluster in lung cancer, what are existing literature or none regarding network analysis in symptoms in lung cancer.

2) General information questionnair -> this is typo, correct please

3) methods: CLARIFY when the symptoms are assessed, after treatments? or durng? or pre treatments?

clarify authors used the SEVERITY of symptoms, or prevalence of symptoms for this network analysis

As this is HUGE different analysis .

4) Gender and smoking network comparision:

authors did not mention about the methods of how to conduct this,

please be specific and provide very details of this information in the method section

Also why gender and smoking are choosen? add background about this subgoup analysis in the INTRODUCTION

Also, add tables of this results including statistical value and p values

5) discussion: please add how to intepret DIFFERENTLY between bridge and central node symptoms. Central node symptoms are most prevalent and severe? ? or just prevalence? or just severe? which dimention of symptoms are used inthis study? Authors clearly should mention the DIFFERENT INTEPRETATION between bridge and central node symptoms.

again, in mentioned in the method section, unclear whether author used SEVERITY or Prevalence (frequency) of ths symptoms.

6) all the figures

I am very confusing

is this based on the severity? or prevalence?

Authors should report BOTH RESULTS OF ALL NETWORK ANALYSIS (FIGURES, AND RESULTS) BASED ON THE SEVERITY AND PREBALENCE.

OR JUST CLEARLY MENTION WHICH SYMPOM DIMENTION WAS CHOSEN FOR THIS MANUSCRIPT AND WHY...

7) Please undergo professional English proofreading service.

Reviewer #2: Providing a brief explanation of "nodal intensity" and "bridge intensity".

The "Conclusion" section could benefit from a more explicit statement about the implications of findings for clinical practice.

**Do you want your identity to be public for this peer review?** For information about this choice, including consent withdrawal, please see our Privacy Policy

Reviewer #1: No

Reviewer #2: No

---

## [Author Response · Author response to Decision Letter 1]

3 Apr 2025

Please refer to the file'Response to Reviewers'.

---

## [Decision Letter · Decision Letter 1]

Contemporaneous symptom networks for multidimensional symptom experience in lung cancer survivors of immunotherapy: a network analysis

PONE-D-24-51139R1

Dear Dr. Yu,

We’re pleased to inform you that your manuscript has been judged scientifically suitable for publication and will be formally accepted for publication once it meets all outstanding technical requirements.

Kind regards,

Saeid Ghavami, PhD

Academic Editor

PLOS ONE

Additional Editor Comments (optional):

Reviewers' comments:

Reviewer's Responses to Questions

**Comments to the Author**

Reviewer #1: All comments have been addressed

2. Is the manuscript technically sound, and do the data support the conclusions?

Reviewer #1: Yes

3. Has the statistical analysis been performed appropriately and rigorously?

Reviewer #1: Yes

4. Have the authors made all data underlying the findings in their manuscript fully available?

Reviewer #1: Yes

5. Is the manuscript presented in an intelligible fashion and written in standard English?

Reviewer #1: Yes

Reviewer #1: well addressed all concerns and comments. I think this paper is ready to be published. Accept is my decision.

**Do you want your identity to be public for this peer review?** For information about this choice, including consent withdrawal, please see our Privacy Policy

Reviewer #1: No

---

## [Editor Report · Acceptance letter]

PONE-D-24-51139R1

PLOS ONE

Dear Dr. Yu,

I'm pleased to inform you that your manuscript has been deemed suitable for publication in PLOS ONE. Congratulations! Your manuscript is now being handed over to our production team.

Kind regards,

on behalf of

Dr. Saeid Ghavami

Academic Editor

PLOS ONE